# Quality of life and experiences of sarcoma trajectories (the QUEST study): protocol for an international observational cohort study on diagnostic pathways of sarcoma patients

Vicky Soomers [1], Ingrid ME Desar,[1] Lonneke V van de Poll-Franse,[2,3,4] Olga Husson,[2,5,6] Winette TA van der Graaf[1,5,7]

OH and WTvdG contributed equally.

For numbered affiliations see end of article.

**Correspondence to**
Dr Olga Husson;
olga.husson@icr.ac.uk

## ABSTRACT

**Introduction** Sarcomas are rare tumours with considerable heterogeneity. Early and accurate diagnosis is important to optimise patient outcomes in terms of local disease control, overall survival (OS) and health-related quality of life (HRQoL). Time to diagnosis is variable in bone as well as soft tissue sarcoma. Possible factors for a long time from first symptom to diagnosis (the total interval) include patient, tumour and healthcare characteristics, but until now the most relevant risk factors and its association with outcomes remain unknown. Our study aims to (1) quantify total interval, the time interval from first symptom until (histological) diagnosis; (2) identify factors associated with interval length and (3) determine the association between total interval and HRQoL, stage and tumour size at diagnosis, progression-free survival (PFS) and OS.

**Methods and analysis** We will conduct a longitudinal, prospective, international, multicentre cohort study among patients aged ≥18 years with newly diagnosed bone or soft tissue sarcoma at eight centres (three in UK, five in The Netherlands). Patients will be asked to complete questionnaires at five points in time; one at diagnosis and at follow-up points of 3, 6, 12 and 24 months. Questionnaire data is collected within the Patient Reported Outcomes Following Initial treatment and Long term Evaluation of Survivorship (PROFILES) registry: an international data management system for collection of patient-reported outcomes. Clinical data will be extracted from patient records. The primary endpoint is HRQoL at diagnosis, measured with the EORTC QLQ-C30. Secondary endpoints are stage and tumour size at diagnosis, PFS, OS, additional patient-reported outcomes, such as quality-adjusted life years and psychological distress.

**Ethics and dissemination** Ethical approval was given by the Health Research Authority and Research Ethics Committee for the United Kingdom (18/WA/0096) and medical ethical committee of Radboudumc for The Netherlands (2017-3881). Results will be presented in peer-reviewed journals and presented at meetings.

**Trial registration number** NCT03441906.

## Strengths and limitations of this study

► The international design allows for comparison of healthcare systems and its influence on total interval length.

► Multicentre, prospective design allows reliable comparison of sarcoma subgroups to make clinically relevant recommendations to improve total interval.

► Inclusion of patients at diagnosis minimises recall bias for total interval length.

► Patients were actively involved in the design of this study and mentioned earlier diagnosis and patient-reported outcomes as research priorities.

## INTRODUCTION

Sarcomas are a group of solid mesenchymal tumours, which comprise more than 70 histological subtypes, with considerable heterogeneity with respect to age at diagnosis, location, biological behaviour and outcome.[1] Approximately 80% of sarcomas are soft tissue sarcomas (STS), the remainder are bone sarcomas. Sarcomas are typical examples of so-called rare cancers, with an estimated European incidence of 4–5 per 100 000 per year when taken all together,[2] accounting for 1% of adult solid malignant cancers.[3] Patients with rare cancers have a higher mortality rate than those with common cancers, due to delays in diagnosis, suboptimal or inadequate treatment, fewer developments in novel therapies and opportunities to participate in clinical trials.[4]

Early and accurate diagnosis of cancer is important to optimise patient outcomes in terms of local disease control, overall survival (OS) and health-related quality of life (HRQoL).[5 6] However, because of the heterogeneity and rarity of sarcomas, there is a lack of public awareness, limited experience

of primary and secondary healthcare professionals and absence of a typical presentation, resulting in late referrals to specialist sarcoma centres and prolonged time to diagnosis.[7]

Time to diagnosis can be defined according to the research framework from Olesen et al,[8] which we adapted to the situation as applicable for sarcomas.[9 10] The time between first symptom and (histological) diagnosis is known as the total interval. This includes a patient and diagnostic interval, defined as time between onset of symptoms until consultation of a healthcare professional and time between consultation of a healthcare professional and diagnosis, respectively. The latter can be further divided into a primary, secondary and tertiary care interval, each of which refers to first consultation until referral to the next caregiver or diagnosis.

Possible risk factors for a prolonged total interval could be patient, tumour or healthcare system characteristics. In order to study the latter, it is informative to compare patients from different countries. In both the Netherlands and UK, general practitioners (GPs) have an important role as healthcare gatekeepers. In general, people consult their GP who then decides whether referral is warranted and determines the acuteness and location of the referral. In the UK, privately insured patients can also self-refer to a hospital without seeing a GP. Furthermore, within the UK, a considerable amount of patients with cancer is diagnosed at an emergency department, associated with worse outcomes.[11] Sarcoma care is formally centralised within the UK, whereas the Netherlands has bone sarcoma centres, and referral to dedicated STS centres is encouraged, but not commissioned. Furthermore, cultural differences may play a role in patient behaviour. Also, longer travel time to a sarcoma centre in the UK compared with the Netherlands may also affect total interval length.

Up to now, only few studies regarding total interval length and clinical outcomes in sarcoma have been published, most were retrospective and included mainly children. Some studies found that a longer total interval worsened OS, while others did not find inferior clinical outcomes.[10] Researchers have argued that this lack of an association, often referred to as the 'waiting-time paradox', may be because the studies have not been able to adequately adjust for the aggressiveness of the cancer tumours. The most significant effect of a long interval for sarcomas seems to be the increasing size of the lesion,[12] with consequent decreased chance of uncomplicated resection with clear surgical margins, a greater risk of amputation and increased risk of developing metastases.[13] This may also affect patient-reported outcomes such as HRQoL of patients with sarcoma.

HRQoL is the patients' perception of his overall health in relation to physical, psychological and social aspects in life.[14] Three systematic reviews have been published on HRQoL of patients with sarcoma, however, none of these looked at the association of total interval length and HRQoL.[15–17] In other cancers and chronic diseases,

lengthening of total interval was associated with decreased HRQoL.[18 19] HRQoL is an interesting outcome parameter for evaluating consequences of long total interval length and provides an insight into the patient's experience of the consequences of diagnostic delay. In addition to using patient-reported outcomes as a measure for quality of care, HRQoL can be used to conduct cost-utility analysis to estimate the ratio between the cost of a prolonged total interval and the benefit of earlier diagnosis in terms of life-years (quality-adjusted life years (QALY)).

Until now, risk factors for a long total interval in adult sarcoma care as well as its effect on clinical and patient-reported outcomes remain unknown. These need to be studied in well-designed, large, prospective studies in order to prioritise interventions to optimise the total interval. Our study aims to quantify total interval, identify independent variables associated with a long interval (such as demographic and clinical factors) and determine the association between total interval and other dependent variables, such as HRQoL, stage and tumour size at diagnosis, progression-free survival (PFS) and OS (figure 1).

## METHODS AND ANALYSIS
### Study design and setting
We will conduct a longitudinal, prospective, cohort study among adult sarcoma patients, newly diagnosed in one of the participating study centres (five centres in the Netherlands: Radboud University Medical Centre Nijmegen, Erasmus Medical Centre Rotterdam, University Medical Centre Leiden, University Medical Centre Groningen, Netherlands Cancer Institute Amsterdam; three centres in the UK: The Royal Marsden London, Christie Manchester, Royal Orthopaedic Hospital Birmingham, all NHS Foundation Trusts). The study started recruitment at the first centre in the Netherlands in February 2018 and in the UK in October 2018 and is currently recruiting.

After informed consent, patients are being asked to complete questionnaires at five points in time: the first at baseline, preferably before start of treatment or within 4 weeks thereafter, and at 3, 6, 12 and 24 months follow-up (table 1). Baseline questionnaire completion will take about 45 min, follow-up questionnaires will take 20–30 min each.

### Patient and public involvement
The different patient-reported outcome measures were selected in consultation with patient advocates. The Sarcoma Patients EuroNet, an international network of patient advocacy groups, has formulated research priorities, at least two of which will be addressed by our study: (1) earlier diagnosis and (2) patient-reported outcomes such as HRQoL.[20] The questionnaire was pilot tested by patients, for acceptability and understandability. Study documents were reviewed by the patients who are members of the Royal Marsden Hospital Patient and Public involvement panel, and the ethics committee

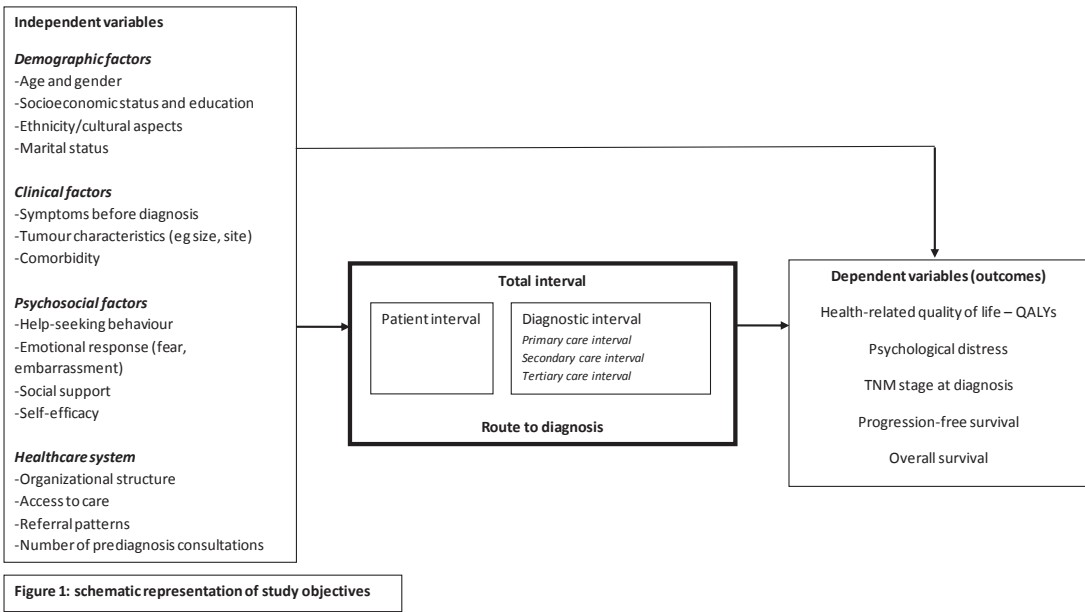

**Figure 1** Conceptual model.

of Radboudumc. The panel and committee provided feedback on the protocol, questionnaires, patient information sheet and informed consent form, regarding content and readability, and changes were incorporated in the final documents. Patients have been and will be involved in study-related presentations and publications.

## Participants

Eligible patients are invited by their treating physician or a member of the research team. Inclusion criteria are: (1) aged ≥18 years; (2) new histological diagnosis of sarcoma as confirmed by a sarcoma histopathologist (according to the International Statistical Classification

| Table 1 Time points and questionnaire items | | | | | | |
|---|---|---|---|---|---|---|
| Item (number of items) | Scale | 0 months | 3 months | 6 months | 12 months | 24 months |
| Characteristics | | | | | | |
| Sociodemographic (max 20) | | X | X | X | X | X |
| Comorbidity (15) | SCQ | X | | | X | X |
| Total interval (max 42) | Own design | X | X | X | X | X |
| Health literacy (1) | SBSQ | X | | | | |
| Social support (1) | QLCS | X | | X | | X |
| Self-efficacy (10) | GSE | X | | | | |
| Coping (28) | Brief COPE | | X | | | |
| Resilience (6) | BRS | | | X | | |
| Outcomes | | | | | | |
| Health-related quality of life (30) | EORTC QLQ-C30 version 3.0 | X | X | X | X | X |
| Quality-adjusted life years (6) | EQ5D5L | X | X | X | X | X |
| Psychological distress (14) | HADS | X | X | X | X | X |
| Financial impact (20) | Own design | X | X | X | X | X |
| Information provision (max 26) | Profiles registry | | X | | | |
| Quality of care (max 23) | PSQ-18 and 3–5 single items | X | X | X | X | X |
| Total number of items | | 194 | 158 | 105 | 113 | 114 |

BRS, brief resilience scale; COPE, Coping Orientation to Problems Experienced; EORTC QLQ, European Organisation for Research and Treatment of Cancer Quality of Life Questionnaire; EQ5D5L, European Quality of life 5 Dimensions 5 Levels; GSE, general self-efficacy; HADS, hospital anxiety and depression scale; PSQ, patient satisfaction questionnaire; QLCS, quality of life-cancer survivors; SBSQ, set of brief screening questions; SCQ, self-administered comorbidity questionnaire.

Of Diseases And Related Health Problems, 10th revision, German Modification (ICD-10-GM) codes C40 and C41 for bone sarcoma and C49 for soft-tissue sarcoma); (3) able to communicate in English or Dutch and to complete questionnaires themselves; (4) mental capacity to provide informed consent and participate in the study (as determined by the healthcare professional) and (5) diagnosed at or referred to one of the participating hospitals. Exclusion criteria are: (1) too ill to complete questionnaires (according to treating physician—patients who experience symptoms are still eligible); (2) desmoid fibromatosis and gastrointestinal stromal tumours due to the different nature of the diseases (ICD-10-GM codes C15-20, C26, C48 and C80).

## Data collection

Eligible patients receive a patient information sheet, which explains the goals and procedure of the study. It includes a link to a secure website (www.profielstudie.nl for both English and Dutch patients), a login name and a password. After logging in, patients can provide informed consent and complete questionnaires online. Patients without access to internet or preference of written communication receive a paper version of the informed consent form and questionnaire. Questionnaires completed on paper will be entered via the data entry option into the Patient Reported Outcomes Following Initial treatment and Long term Evaluation of Survivorship system (PROFILES; www.profilesregistry.nl[21]) by a member of the study team. The data entry portal has the same format as the online questionnaire data, minimising the chance of errors and enhancing data extraction. Paper questionnaires will be stored in a secured room at study coordinating sites (Radboudumc and Royal Marsden Hospital). PROFILES is a data management system set up in 2009 in the Netherlands for the study of physical and psychosocial impact of cancer and its treatment. The data collected in PROFILES is stored on a secure server in the Netherlands. In order to retrieve the data, an authorised member of the study team can login and download an SPSS or Excel file containing the encoded questionnaire data. PROFILES has been developed to the requirements of the higher education and research community and allows end-to-end encryption.

The research coordinator has access to a password-protected file that links patients' study numbers to their electronic patient record number. Clinical data and survival data will be retrieved from the patients' medical record by a member of the study team into the electronic case report forms (eCRF) database (MACRO), which is maintained according to current norms and International Council for Harmonisation of Technical Requirements for Registration of Pharmaceuticals for Human Use - Good Clinical Practice (ICH-GCP)) standards and is password protected. Patient records will not leave the hospital.

Finally, questionnaire data will be linked with the eCRF database (all encoded data) using study numbers. The combined dataset will be stored under appropriate password protection. Data will be recorded and retained in accordance with the Data Protection Act 1998.

## Case report forms

CRFs will be completed at five time points during the study. The first will be completed on inclusion, the following time points coincide with the completion of follow-up questionnaires. The last CRF is also the end-of-study CRF, which can be completed before 24 months if a patient withdraws or deceases. The information collected on the CRF will be stored on a secure CRF database using anonymous study numbers. Data collected includes documentation of eligibility criteria, date of diagnosis, tumour characteristics such as histology, tumour, node and metastasis (TNM) stage, tumour size, treatment regiment, re-occurrence of disease or metastases, reason for withdrawal of the study and time of death, if applicable.

## Questionnaires

We have combined self-designed questions and several validated questionnaires designed by other researchers (details below). For non-commercial scientific use, no formal licenses are needed for the use of these questionnaires. Self-designed items and existent questionnaires not available in both English and Dutch were translated with formal forward-backward translation by bilingual speakers. Table 1 summarises the time points at which each construct is being measured.

## Sociodemographics

The questionnaires contain questions on sociodemographic characteristics of the participant at the time of questionnaire completion, such as marital status and educational level. Comorbidity is being measured with the Self-administered Comorbidity Questionnaire (SCQ), which is a validated list where patients report their comorbidity during the past year.[22]

## Total interval

A 42-item list was self-designed to assess the total interval. Examples of questions are as follows: 'With which symptom(s), caused by the sarcoma, did you first go to a doctor?', 'To which doctor did you first talk about your symptoms?' and 'How often did you talk to the following doctors about your symptoms belonging to sarcoma, before you heard you had a sarcoma?' At follow-up, a few questions are repeated to complete data collection. We will sample survey the reported dates by cross-checking them with the patient's record. If more than 5% of the cross-checked dates deviates more than 1 month from the registered dates in the medical record, we will cross-check all dates and use the clinical reported dates for statistical analysis.

Health literacy is being assessed by a Dutch adaptation of Chew's Set of Brief Screening Questions (SBSQ) in a single-item question.[23–25]

Social support is being assessed by one single item: 'Was the amount of support you received from others

sufficient?' extracted from the Quality of Life-Cancer Survivors questionnaire.[26 27]

Self-efficacy is measured with the General Self-Efficacy scale (GSE).[28] This 10-item scale assesses a general sense of perceived self-efficacy with the aim to predict coping with daily hassles as well as adaptation after experiencing a stressful life event. Self-efficacy is the belief that one can perform a novel or difficult task, or cope with adversity. Perceived self-efficacy facilitates goal-setting, effort investment, persistence in face of barriers and recovery from setbacks. Responses are made on a 4-point scale. A higher final composite score correlates with higher perceived self-efficacy.

Coping is assessed in the 3-month questionnaire with the help of the brief Coping Orientation to Problems Experienced (COPE).[29] Coping is about emotional and mental reactions, which enable people to activate sources of help needed to cope with stress and problems. This 28-item scale measures 14 positive and negative styles of coping on a 5-point Likert-scale.

Resilience is measured in the 6-month questionnaire using the Brief Resilience Scale (BRS).[30] Resilience is a skill that helps people recover from a life event. People with high (perceived) resilience can move on faster after a setback. The BRS is a 6-item scale with a 5-point Likert scale.

### Health-related quality of life
HRQoL is being assessed with the EORTC QLQ-C30, V.3.0, which is validated and available in English and Dutch.[31] This 30-item HRQoL questionnaire consists of five functional scales (physical, role, cognitive, emotional and social), a global quality of life scale, three symptom scales (fatigue, pain, nausea and vomiting) and a number of single items assessing common symptoms (dyspnoea, loss of appetite, sleep disturbance, constipation and diarrhoea) and perceived financial impact of the disease. After linear transformation, all scales and single-item measures have scores ranging from 0 to 100. A higher score on the functional scales and global QoL means better functioning and HRQoL, whereas a higher score on the symptom scales means more complaints.

### Quality-adjusted life years
QALY is being measured with the EuroQol EQ-5D-5L, which is a descriptive system for the measurement of health.[32] It measures HRQoL on five dimensions: mobility, self-care, usual activities, pain-discomfort and anxiety-depression. To make the EQ-5D-5L suitable for use in economic evaluations, the health status needs to be valued with a preference-elicitation method.[33] Both Dutch and English national values were collected and subsequently modelled.[34 35]

### Psychological distress
Psychological distress is being assessed with the Hospital Anxiety and Depression Scale (HADS), which is validated in Dutch and English.[36] This 14-item instrument measures psychological distress, with seven items each assessing anxiety and depression. The summed total score of the HADS will be used to reflect psychological distress. Higher total scores are indicative of more psychological distress.

### Financial impact
We self-designed a 20-item questionnaire regarding financial barriers to care. The questions were designed based on a literature study of items that are important in health-seeking behaviour but have not been validated. Topics covered are financial barriers to care, financial impact of living with cancer, personal expenses and potential solutions for reducing financial impacts.

### Information provision
Five self-designed questions with multiple items are being asked to identify time points and subjects on which participants would like more information.

### Quality of care
Quality of care is being assessed with the 18-item Patient Satisfaction Questionnaire (PSQ-18),[37] available in both English and Dutch.[38 39] This instrument yields scores for each of the seven different subscales: general satisfaction, technical quality, interpersonal manner, communication, financial aspects, time spent with doctor and accessibility and convenience. High scores reflect satisfaction with medical care. In addition, three to five self-designed single items to assess overall satisfaction of care at the primary doctor's office, hospital and sarcoma centre are being asked.

### Endpoints
The primary endpoint is HRQoL of patients with sarcoma at diagnosis (baseline) as measured with the EORTC QLQ-C30 (global health status). Secondary endpoints are: QALY, psychological distress, stage and tumour size at diagnosis, PFS and OS.

If subgroups are large enough, we will conduct these analyses for different clinically relevant subgroups, such as different histological subtypes, geographical areas, and so on.

### Sample size calculation
We expect a minimum response rate at baseline of 65%, based on rates in other PROFILES studies.[40] During follow-up, after completion of the first questionnaire, we expect a response rate of 80%. The definition of a long total interval will follow from our statistical analysis (see below), however, if the analysis does not provide a clear cut-off point, we will use the last quartile to define the population with a long total interval.

Using the EORTC QLQ-C30, differences of at least 10 points have been considered as clinically meaningful.[41] Based on results from our ongoing PROFILES studies, an SD of about 20 points for each scale can be expected. Using an alpha of 0.05, a power of 0.90 and a long diagnostic interval of 25% in the total group of patients with

sarcoma, with the expected drop-out, would require 265 patients.[42] In order to make country-to-country comparisons, we aim to include 265 Dutch and 265 English patients in a time frame of 18 months with a total follow-up of 24 months.

## Statistical analysis

Descriptive statistics (means, SD, median, range, frequencies) will be used to quantify diagnostic intervals and describe the study population.

HRQoL at baseline will be calculated according to the EORTC scoring manual.[43] Missing items will be imputed according to these guidelines, after which an available case analysis will be performed.

The relationship between total interval length and HRQoL at baseline will be investigated by plotting HRQoL against total interval length as a continuous variable. Linear regression will be used to assess their association. The time point providing a significant difference in HRQoL will be used as a cut-off point for further analysis. If this does not provide a clear cut-off point, logistic regression will be used to assess an association between baseline HRQoL and total interval grouped into suitable categories, such as quartiles. The last quartile will then be used to define the population with a long interval.

Apart from statistical significance, we will look at clinically relevant differences in HRQoL scores as determined by Cocks *et al.*[42] A small effect size will then be considered as an appropriate value for a cut-off point.

A series of univariate logistic regression analyses will be conducted to assess the relationship between total interval length (grouped by the cut-off point as defined by the previous analysis) and independent variables, such as patient, tumour and healthcare system risk factors. All factors with $p<0.1$ will then be used in multiple logistic regression analysis (forced entry method) to investigate whether these factors are independently associated with total interval length.

Apart from total interval length, the association of other patient and tumour characteristics (such as self-efficacy, social support, financial difficulties, histology), and HRQoL at baseline will be investigated using univariate logistic regression analysis. Using the forced entry method, multiple logistic regression analysis will then be performed with all factors with $p<0.1$ to assess what factors are independently associated with baseline HRQoL.

Change in HRQoL during the follow-up period of 2 years and factors associated with changes in HRQoL will be analysed using repeated measures mixed models. This will be compared between patients with a short and long total interval, using repeated measures analysis of variance, controlling for relevant patient and tumour characteristics, and the patient's baseline score. Clinically relevant differences will be assessed using Cocks' method.[41 42]

Other patient-reported outcomes such as QALYs and psychological distress will be analysed in the same way.

Multivariate analyses will be performed to examine associations between total interval length and (1) QALYs, (2) psychological distress, (3) stage at diagnosis and (4) tumour size. These analyses will be corrected for potential confounders including patient and tumour characteristics and healthcare system.

Both unadjusted and adjusted multivariate Cox proportional hazard regression analyses will be used to examine whether a long total interval is associated with PFS or OS.

PFS is defined as the time interval between diagnosis until clinical or radiological progression, as assessed by the treating consultant. OS is defined as the time from diagnosis until death.

Statistical analyses will be performed using IBM SPSS V.25.0; two-sided p values $<0.05$ will be considered statistically significant.

## Missing data

Online questionnaire completion does not allow for missing data, unless participants have not completed the entire questionnaire, as patients are unable to proceed to the next question until all questions on the current page have been answered. Items missing from paper questionnaires will be dealt with as missing at random. The EORTC QLQ-C30 allows imputation of missing values according to the EORTC scoring manual guideline.[43] Numbers of missing items will be reported.

## Impact of COVID-19 pandemic on QUEST

A national lockdown was introduced across The Netherlands on 16 March and the UK on 23 March 2020, as part of the national strategies to flatten the curve of the COVID-19 pandemic. On 23 March, recruitment for QUEST was finished in The Netherlands, while the recruitment target was almost reached in the UK. The COVID-19 pandemic forced us to put the recruitment on hold in the UK. The negative consequences of the pandemic on cancer diagnostic timelines (prolonged), incidence (reduced) and eventually cancer outcomes have been shown and modelled by several studies.[44–46] We will therefore discuss the necessity to reopen recruitment in the UK with our statistical department, as patients recruited during the pandemic will not be representative for the sarcoma population outside COVID-19 times and will bias our results.

**Author affiliations**

[1]Department of Medical Oncology, Radboud University Medical Centre, Nijmegen, The Netherlands
[2]Division of Psychosocial Research and Epidemiology, The Netherlands Cancer Institute, Amsterdam, The Netherlands
[3]Department of Research, Netherlands Comprehensive Cancer organization (IKNL), Utrecht, The Netherlands
[4]Department of Medical and Clinical Psychology, CoRPS – Centre of Research on Psychology in Somatic Diseases, Tilburg University, Tilburg, The Netherlands
[5]Department of Medical Oncology, Netherlands Cancer Institute, Amsterdam, The Netherlands
[6]Division of Clinical Studies, Institute of Cancer Research, London, UK
[7]Department of medical oncology, Royal Marsden NHS Foundation Trust, London, UK

**Acknowledgements** We thank the patients who contributed to the design of this study.

**Contributors** OH and WTAvdG were involved in conceptualisation. VS, LVvdP-F, IMED, OH and WTAvdG designed the study. VS, OH and WTAvdG were responsible for writing the protocol. VS and OH were responsible for obtaining ethical approval for the study. LVvdP-F and OH were responsible for the use of the PROFILES registry. VS, OH and WTAvdG were involved in data collection and analysis, and all authors in data interpretation. All authors read and approved the final manuscript.

**Funding** VS (clinical research fellow) is funded by a research grant from the Radboud Institute of Health Sciences at Radboudumc Nijmegen, the Netherlands. OH is supported by a Social Psychology Fellowship from the Dutch Cancer Society (#KUN2015-7527) and and Netherlands Organization for Scientific Research VIDI grant (198.007). This study represents independent research supported by the NIHR Biomedical Research Centre at The Royal Marsden NHS Foundation Trust and the Institute of Cancer Research London (grant number: NA). The PROFILES registry was funded by an Investment Grant (#480-08-009) of the Netherlands Organization for Scientific Research (The Hague, The Netherlands). Within the UK, the study has received NIHR CRN support (grant number: NA). The sponsor for this study is the Institute of Cancer Research. The sponsor contact is Dr Barbara Pittam (Barbara. pittam@icr.ac.uk). Role of sponsor: Trial insurance and indemnity, scientific and statistical review of the protocol andstudy documents, ethical approval and protocol amendments, maintaining study master file, ensuring data and documentation are available for monitoring, inspection or audit, case report design, data analysis, preparation and dissemination of results (posters, presentations and publication), data storage.

**Disclaimer** The views expressed are those of the authors and not necessarily those of the NIHR or the Department of Health and Social Care.

**Competing interests** None declared.

**Patient and public involvement** Patients and/or the public were involved in the design, or conduct, or reporting, or dissemination plans of this research. Refer to the Methods section for further details.

**Patient consent for publication** Not required.

**Ethics and dissemination** The study was approved by the Health Research Authority and Research Ethics Committee of the United Kingdom (18/WA/0096), and by the medical ethical committee of Radboudumc for The Netherlands (2017-3881). Under Dutch law, approval for observational and questionnaire research by one medical ethical committee is sufficient to implement the study at multiple Dutch centres. The study was registered at clinicaltrials.gov (NCT03441906). Results from the QUEST study will be published and disseminated via peer-reviewed journals, local, national and international conferences and via patient meetings and patient advocates.

**Provenance and peer review** Not commissioned; externally peer reviewed.

**ORCID iD**
Vicky Soomers http://orcid.org/0000-0003-0019-3968

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
