## [Reviewer comments · BMJ Open]

ARTICLE DETAILS

TITLE (PROVISIONAL)	QUality of life and Experiences of Sarcoma Trajectories (the QUEST study): protocol for an international observational cohort study on diagnostic pathways of sarcoma patients
AUTHORS	Soomers, Vicky; Desar, Ingrid; van de Poll-Franse, Lonneke; Husson, Olga; van der Graaf, Winette

VERSION 1 – REVIEW

REVIEWER	Juho Salo Helsinki University Hospital, Helsinki, Finland
REVIEW RETURNED	04-Jun-2020

GENERAL COMMENTS	Dear authors, it's my pleasure as sarcoma surgeon to review this study protocol. I'm waiting your results. You have quite ambitious study protocol, which requires very good co-operation for the patients. Best luck.
---

REVIEWER	Gaurav Goyal University of Alabama at Birmingham
REVIEW RETURNED	21-Jun-2020

GENERAL COMMENTS	The authors have submitted a protocol for an international observational cohort study to study the association of interval from symptom onset to diagnosis of sarcoma with quality of life and other outcomes measures like PFS and OS. The study is clearly presented, and will lead to important data in the field. I have couple of questions/comments: 1. I don't see the date of completion of study mentioned. Please include it in abstract, manuscript text, as well as the figure of design.2. The authors mention in exclusion criteria - patient being too ill to participate in the study according to physician. I think one of the goals of the study was to see if prolonged interval to diagnosis was associated with worse QoL, PFS, and OS. If they exclude patients based on physician choice of "illness" status, they may lose a very important group of patients from their study. I would argue that the ones most "ill" may be the ones you definitely want to include in the study. In real practice, patients who are too sick might themselves not want to answer questionnaire. However, to make the study pragmatic and to capture the really sick patients (with worse QoL, PFS, OS, and arguably could be the ones with long intervals to diagnosis), the investigators could consider a simple interview or even the question of "when did symptoms start" and correlate with simple QoL tools or just PFS and OS. Nevertheless,
---

	it would be important to simply mention in results of the eventual manuscript regarding how many patients met this exclusion criteria for full transparency. 3. Please consider providing the self-generated questionnaire as an appendix to foster education and collaborations.
--	--

REVIEWER	Kirsten Ness St. Jude Children's Research Hospital
REVIEW RETURNED	06-Aug-2020

GENERAL COMMENTS	This is a well written description of the study as designed. It would be timely if the authors indicated how they plan to continue enrollment during the pandemic and if they have made any modifications to the protocol.
--

VERSION 1 – AUTHOR RESPONSE

Reviewer#1: Dear authors, it's my pleasure as sarcoma surgeon to review this study protocol. I'm waiting your results. You have quite ambitious study protocol, which requires very good co-operation for the patients. Best luck.

Response: We thank the reviewer for the compliments.

Reviewer#2: The authors have submitted a protocol for an international observational cohort study to study the association of interval from symptom onset to diagnosis of sarcoma with quality of life and other outcomes measures like PFS and OS. The study is clearly presented, and will lead to important data in the field.

Response: We thank the reviewer for the compliments.

Reviewer#2: I don't see the date of completion of study mentioned. Please include it in abstract, manuscript text, as well as the figure of design.

Response: We thank the reviewer for the thorough review. At the time of submission the study was not completed. One of the requirements of BMJ Open is that the study is not finished at the time of submission (<https://bmjopen.bmj.com/pages/authors/#protocol>). We do understand however that it is important to get insight into the total study duration and therefore we have added the expected recruitment time to our body text as requested on page 10: "In order to make country-to-country comparisons, we aim to include 265 Dutch and 265 English patients in a timeframe of 18 months with a total follow-up of 24 months."

We have decided not to include this recruitment time in the abstract as the start date is also not mentioned there. The figure only shows the conceptual design of the study and no timelines.

Reviewer#2: The authors mention in exclusion criteria - patient being too ill to participate in the study according to physician. I think one of the goals of the study was to see if prolonged interval to diagnosis was associated with worse QoL, PFS, and OS. If they exclude patients based on physician choice of "illness" status, they may lose a very important group of patients from their study. I would argue that the ones most "ill" may be the ones you definitely want to include in the study. In real

practice, patients who are too sick might themselves not want to answer questionnaire. However, to make the study pragmatic and to capture the really sick patients (with worse QoL, PFS, OS, and arguably could be the ones with long intervals to diagnosis), the investigators could consider a simple interview or even the question of "when did symptoms start" and correlate with simple QoL tools or just PFS and OS. Nevertheless, it would be important to simply mention in results of the eventual manuscript regarding how many patients met this exclusion criteria for full transparency.

Response: We agree with the reviewer that the "sick" patients are of utmost importance to monitor. We only exclude those patients who are too ill to complete a questionnaire by themselves. In practice this will only be a very limited number of patients. Patients who experience symptoms of disease are included in our study. We have clarified this in our methods section on page 6: "too ill to complete questionnaires (according to treating physician e.g. those who are terminally ill; patients who experience symptoms of disease are still eligible)"

We decided not to use proxies for HRQoL assessment as there is a bunch of literature available that indicates low reliability of these assessments, especially for concepts measuring more subjective domains (e.g. <https://link.springer.com/article/10.1023/A:1013187903591>).

We do agree that it is important to know how many patients met this exclusion criterium, however at the time of first submission the recruitment of the study was not finished.

Reviewer#2: Please consider providing the self-generated questionnaire as an appendix to foster education and collaborations.

Response: We do agree with the reviewer that publication of the questionnaires would be highly informative for the readers, however several of the questionnaires were developed and validated by other research groups and already published / a fee has to be paid to be published. We therefore decided not to add the questionnaire to the appendix.

Reviewer#3: This is a well written description of the study as designed. It would be timely if the authors indicated how they plan to continue enrollment during the pandemic and if they have made any modifications to the protocol.

Response: We thank the reviewer for this comment. At the time the pandemic started we finished recruitment in The Netherlands and almost finished recruitment in UK. We are currently discussing with a statistician if our sample size is large enough to close recruitment, as several studies have shown the negative impact of the pandemic on e.g. cancer incidence (reduced) and diagnostic timeliness (prolonged) and cancer outcomes. For example:

[https://www.thelancet.com/journals/lanonc/article/PIIS1470-2045\(20\)30388-0/fulltext](https://www.thelancet.com/journals/lanonc/article/PIIS1470-2045(20)30388-0/fulltext)

We have added a paragraph about the influence of the pandemic to our discussion on page 11:

"Impact of COVID-19 pandemic on QUEST

A national lockdown was introduced across The Netherlands on March 16 and the UK on March 23, 2020, as part of the national strategies to flatten the curve of the COVID-19 pandemic. On March 23 recruitment for QUEST was finished in The Netherlands, while the recruitment target was almost reached in the UK. The COVID-19 pandemic forced us to put the recruitment on hold in the UK. The negative consequences of the pandemic on cancer diagnostic timelines (prolonged), incidence

(reduced) and eventually cancer outcomes has been shown and modelled by several studies [44-46]. We will therefore discuss the necessity to reopen recruitment in the UK with our statistical department, as patients recruited during the pandemic will not be representative for the sarcoma population outside COVID times and will bias our results.”

44. Dinmohamed, A.G., et al., Fewer cancer diagnoses during the COVID-19 epidemic in the Netherlands. *Lancet Oncol*, 2020. 21(6): p. 750-751.

45. Maringe, C., et al., The impact of the COVID-19 pandemic on cancer deaths due to delays in diagnosis in England, UK: a national, population-based, modelling study. *Lancet Oncol*, 2020. 21(8): p. 1023-1034.

46. Sud, A., et al., Effect of delays in the 2-week-wait cancer referral pathway during the COVID-19 pandemic on cancer survival in the UK: a modelling study. *Lancet Oncol*, 2020. 21(8): p. 1035-1044.

VERSION 2 – REVIEW

REVIEWER	Gaurav Goyal University of Alabama at Birmingham
REVIEW RETURNED	08-Sep-2020

GENERAL COMMENTS	Thank you for answering the questions. I look forward to seeing the results of your study.
--

REVIEWER	Kirsten Ness St. Jude Children's Research Hospital
REVIEW RETURNED	20-Sep-2020

GENERAL COMMENTS	This is a well written description of this clinical trial with adequate references to the difficulties the study may encounter because of the pandemic. The study looks at important information for a rare disease which may be able to inform care, and the health systems in both these countries in the future.
---